# Overlapping Stent Treatment for Ruptured Dissecting Aneurysms in Posterior Circulation

**DOI:** 10.3390/brainsci13111507

**Published:** 2023-10-25

**Authors:** Minghui Zhou, Zengbao Wu, Ali Abdi Maalim, Ying Zeng, Xiao Guo, Zhenhua Zhang, Xiaohong Yuan, Zacharia Majaliwa Enos, Kai Shu, Ting Lei, Mingxin Zhu

**Affiliations:** 1Department of Neurosurgery, Tongji Hospital, Tongji Medical College, Huazhong University of Science and Technology, Wuhan 430030, China; zhouminghui13@163.com (M.Z.); izengbao@tjh.tjmu.edu.cn (Z.W.); alimaalim1@hotmail.com (A.A.M.); tuzi76@126.com (Y.Z.); 13886404502@163.com (X.G.); zhangzhenhua197603@163.com (Z.Z.); 15071381712@163.com (X.Y.); kshu@tjh.tjmu.edu.cn (K.S.); tlei@tjh.tjmu.edu.cn (T.L.); 2Department of Neurosurgery, Tongji Tianyou Hospital Affiliated to Wuhan University of Science and Technology, Wuhan 430030, China; ezacharia30@gmail.com

**Keywords:** overlapping stents, ruptured dissecting aneurysms, posterior circulation, endovascular treatment

## Abstract

Ruptured dissecting aneurysms in posterior intracranial circulation present significant clinical challenges and often cause poor prognoses. Our cohort used overlapping stents as the primary treatment. We analyzed the medical records of 27 patients (18 men/nine women) with ruptured posterior circulation dissecting aneurysms (PCDAs). Their average age was 52 years. We selected 11 patients who used Enterprise (EP) and LVIS stents overlappingly and matched them 1:1 with counterparts who received either EP or LVIS stents individually. Overlapping stents was a feasible treatment in all 27 cases. We successfully followed up 26 patients for ≥6 months. Regrettably, one patient died from intracranial hypertension on Day 7 post-procedure. Immediate post-procedure angiographies indicated Raymond grade I, II, and III occlusions of PCDAs in 16 (59.3%), 7 (25.9%), and 4 (14.8%) cases, respectively. At an average follow-up duration of 16.2 months, 25 patients (96.2%) had modified Rankin Scale scores of 0–2, signifying positive outcomes. One patient (3.8%) had a score of 3–4. Recurrence rates for the EP and LVIS stent groups were higher than those of the overlapping stent group (45.45% vs. 9.09%, *p* = 0.15 and 27.27% vs. 9.09%, *p* = 0.59, respectively). No significant difference in recurrence rates existed between the overlapping and single-stent groups. Similarly, follow-up outcomes were consistent between the two groups. Overlapping stents could be an efficient method for treating ruptured PCDAs.

## 1. Introduction

Dissecting aneurysms in the posterior intracranial circulation pose treatment difficulties due to their association with high morbidity and mortality [1,2,3]. Ruptured posterior circulation dissecting aneurysms (PCDAs) have an early re-bleeding rate exceeding 70%, often resulting in a grim prognosis [4,5]. Previous studies have shown that hemodynamics and morphological factors, such as the aspect ratio, height width ratio, and size ratio, play equally important roles in the rupture process of intracranial saccular aneurysms (ISA) [6,7]. However, there are numerous contradictions in these studies regarding the prediction of rupture risk in ISA [8,9,10,11,12]. These contradictory results are caused by the pathophysiological complexity of ISA [6]. Compared to ISA, PCDAs have a different pathophysiology and more complex hemodynamics, leading to ambiguities and unknowns in diagnosis and treatment [13,14,15,16,17]. Recognizing clinical manifestations and prioritizing early treatment are vital in determining strategic options, encompassing both aggressive surgical and individualized endovascular therapy. Surgical treatment of PCDAs often faces complications due to its deep location, as well as the potential for cranial nerve and brainstem injury. Conversely, endovascular management offers a less invasive alternative, presenting potential advantages over open surgery [3,18].

Endovascular treatment encompasses both proximal occlusion and reconstruction techniques. However, the ideal endovascular treatment strategy for these aneurysms remains unclear. Endovascular trapping or sacrificing the patent artery is a practical approach to prevent re-rupture. Yet, if the diseased segment extends into the basilar artery (BA) or vertebral artery (VA) on the dominant blood supply side, sacrificing the parent vessel could result in a high risk of ischemic stroke, leading to a poorer prognosis [19,20]. Reconstructing the parent artery using a stent might be a safer and more feasible management solution, helping to preserve the parent artery [21]. Flow diverter (FD) devices could reduce recurrence rates, especially, benefiting large or complex aneurysms more than traditional stents. However, these devices also present a significant risk of branch occlusion in the posterior circulation ruptured aneurysms, potentially leading to severe complications [22,23].

Recently, the use of overlapping stents has been explored for treating PCDAs, demonstrating favorable conditions for enhancing blood thrombosis [24,25,26]. To the best of our knowledge, there are limited studies on treating PCDAs using overlapping stents in acute hemorrhagic scenarios. In our study, we address this gap by presenting the outcomes and prognosis of 27 PCDA patients, including those with aneurysms affecting the Vertebral Artery (VA), Basilar Artery (BA), Posterior Cerebral Artery (PCA), and Posterior Inferior Cerebral Artery (PICA). We hypothesize that using overlapping stents for treating PCDAs serves a dual purpose: The primary laser-cut stent ensures complete embolization, helping prevent secondary rupture of the aneurysm. The subsequent deployment of a second braided stent enhances metal coverage, contributing significantly to vascular reconstruction. This refined approach boosts safety and markedly enhances effectiveness in managing ruptured dissecting aneurysms within the posterior circulation.

## 2. Materials and Methods

### 2.1. Inclusion and Exclusion Criteria

Figure 1 illustrates the enrollment procedure of our study for intracerebral aneurysm patients. After meeting specific criteria, we included patients who (1) had experienced an aneurysmal subarachnoid hemorrhage, confirmed with a CT scan; (2) had a posterior circulation dissecting aneurysm (PCDA) confirmed via digital subtraction angiography (DSA) or computed tomographic angiography (CTA); (3) underwent overlapping stents-assisted coiling treatment; and (4) were treated at Tongji Hospital of Huazhong University of Science and Technology and followed up for a minimum of six months. Patients not meeting these criteria were excluded from our study.

### 2.2. Patients and Study Design

This study was approved by the Institutional Review Board of Tongji Hospital, affiliated with Huazhong University of Science and Technology. We performed a retrospective analysis of departmental and hospital databases to identify all patients diagnosed with intracerebral aneurysms between November 2018 and October 2021. Complete radiological and clinical records for this period were accessible from all databases. We described the clinical presentation, radiologic features, endovascular procedure, and clinical outcomes of 27 consecutive patients treated using overlapping stent placement.

### 2.3. Clinical Management

We recorded patient clinical data and demographics, including the Hunt–Hess grade at presentation and the Fisher grade on initial non-contrast head CT. Patients were admitted to the neurosurgery intensive care unit (NICU), and necessary supportive care was provided. A multidisciplinary team, comprising cerebrovascular surgeons and interventional neuroradiologists, determined the treatment approach. We recorded extensive clinical data, including pre-operation and post-operation notes, radiologic images, neuroradiologic reports, any re-rupture before treatment, and complications of SAH experienced before treatment. We also analyzed the baseline clinical data and aneurysm characteristics.

### 2.4. Endovascular Procedure

Procedures were executed under general anesthesia using the transfemoral approach. The detailed process involved the placement of a 6.0 Fr Envoy guiding catheter in the distal VA (6 Fr; Envoy, Cordis, FL, USA). A stent-line microcatheter was navigated to a distal branch of the diseased segment (Headway 21, MicroVention, Aliso Viejo, CA, USA; Excelsior XT-27, Stryker, Kalamazoo, MI, USA; Infusion Catheter, Cordis, FL, USA, or Excelsior SL-10, Stryker, USA). A coil-line microcatheter was inserted into the aneurysm sac (Headway 17, MicroVention, USA, or Echelon-10, Covidien/ev3, Menlo Park, CA, USA). The aneurysm was embolized with detachable coils (MicroVention, USA; Stryker, USA). The first stent (Enterprise, Codman, Raynham, MA, USA; Neuroform EZ or Altas, Stryker, Kalamazoo, MI, USA) was deployed with coils as a general procedure. Then, the second stent (LVIS, MicroVention, Aliso Viejo, CA, USA; Leo or Leo’s Baby, Balt, rue de la Croix Vigneron, Montmorency, France) was transported using a microcatheter to overlap the neck and reconstruct the parent artery. All patients were admitted to the hospital for subarachnoid hemorrhage. No antiplatelet or anticoagulation was administered pre-operation, but 0.1 µg/kg/min Tirofiban was administered during the first stent’s deployment. Post-procedure, all patients were prescribed dual antiplatelet therapy consisting of aspirin (100 mg/day) and clopidogrel (75 mg/day) for 6 weeks and then aspirin (100 mg/day) only for another 6 months.

### 2.5. Evaluation and Follow-Up

We conducted follow-up assessments for all patients through telephone consultations to evaluate their overall well-being. Additionally, they were scheduled for in-hospital examinations six months after the operation for a comprehensive assessment. Clinical outcomes were assessed using the Modified Rankin Scale (mRS) to categorize patient outcomes as follows: good (mRS score of 0–2), moderate (mRS score of 3–4), or poor (mRS score of 5–6). The percentage of aneurysm sac obliteration was determined using the modified Roy–Raymond scale (MRRC) by DSA in all patients: (1) complete occlusion was defined as a Raymond Grade I; (2) the presence of a neck remnant and residual sac was defined as a Raymond Grade II; and (3) contrast filling in the aneurysmal neck, along with a cavity, was defined as Raymond Grade III. All patients were initially admitted to the hospital for subarachnoid hemorrhage; no antiplatelet or anticoagulation medications were administered before the procedure. Following treatment, patients were advised to continue daily dual antiplatelet medication for three months, along with 100 mg of aspirin for the subsequent six months. Among the 27 patients who underwent overlapping stent procedures, we selected 11 patients who utilized Enterprise (EP) and LVIS stents in an overlapping manner. They were matched in a 1:1 ratio with those who received either EP or LVIS stents separately. Two experienced cerebrovascular surgeons independently conducted radiological and clinical assessments for the follow-up evaluations.

### 2.6. Statistical Analysis

Statistical analyses were performed using SPSS version 21. Data were presented as number (percentage) or mean ± standard deviation (SD). We used the chi-square test and the t-test for comparisons. A *p*-value of < 0.05 was considered statistically significant.

## 3. Results

All aneurysms were successfully treated on the first attempt. Table 1 summarizes patient demographics and aneurysm baseline characteristics. The average age was 52.3 years (range, 9–77 years). The percentage distribution of ruptured PCDAs by location was as follows: VA (*n* = 14, 51.2%), BA (*n* = 8, 29.6%), PCA (*n* = 3, 11.1%), and PICA (*n* = 2, 7.4%). At presentation, 33.3% (9/27) of patients were Hunt–Hess grade 1, 44.4% (13/27) were grades 2–3, and 5 patients (18.5%) were grades 4–5. No aneurysms re-ruptured during admission or endovascular management, which, in all cases, took place within 48 h of clinical presentation. Seven patients (25.9%) with moderate-to-severe hydrocephalus underwent external ventricular drain placement. A follow-up was successful for 26 patients over 6–37 months, with an average follow-up time of 16.2 months.

All stents fully opened under fluoroscopy in the desired parent artery. The average dissection size was 10.6 mm, with eight being large and three giant aneurysms. Major complications included cerebral edema (18.5%), hydrocephalus (25.9%), and symptomatic procedure-related stroke (7.4%). No patient had a myocardial infarction or gastrointestinal bleeding. There were no deaths during the periprocedural period. In one case, a patient initially classified as Hunt–Hess grade 5 before the procedure showed no improvement in intracranial hypertension post-procedure. The family declined further surgical intervention, opting for hospice care instead. The patient died on the seventh post-operative day. All other patients had at least two follow-up clinic visits. One patient with a complex ruptured left VA dissecting aneurysm underwent Enterprise overlapping LVIS stent-assisted coil embolization and achieved complete obliteration, as seen 23 months post-procedure (Figure 2). In two cases, patients had ruptured dissecting aneurysms in the left PCA and were treated with overlapping double stents. Twelve months after treatment, the aneurysms were stable and showed Roy–Raymond grade embolization (Figure 3). The other patients who had ruptured dissecting aneurysms in the PCA, BA, and VA were also treated with overlapping stent-assisted coil embolization and complete obliteration (Figure 4). The clinical characteristics, overlapping stent types, and complications of the patients are detailed in Table 2.

Overlapping stents were successfully deployed in all patients. Among the patients, two experienced cerebral infarcts due to the procedure, with one patient displaying left-side hemiparesis after the operation. In the other 24 patients, no new permanent neurologic deficits from hemorrhagic or ischemic stroke occurred within 4 weeks post-procedure. Follow-up DSA examination revealed that all cases of ruptured PCDAs were completely occluded.

By contrast, only one case involving the posterior cerebral artery (PCA) segment was completely obstructed, resulting in the disappearance of the dissecting aneurysm. A follow-up catheter angiography was conducted for 26 patients. Nearly all patients regained independence in their activities and resumed work and daily routines (96.2%). Among the dissecting aneurysms, 24 (92.3%) had achieved full occlusion, while 2 (7.7%) showed remnants at the last follow-up. Clinical characteristics and individual angiographic outcomes are detailed in Table 3. Within the group that received overlapping stents, there was a singular instance (9.09%) of aneurysm recurrence occurring approximately 22.91 ± 3.62 months after the initial procedure (refer to Figure 5). Subsequently, this patient underwent another endovascular treatment. We conducted a rigorous statistical analysis to ensure the accuracy of our findings. When comparing recurrence rates, we observed that the overlapping stents group had a lower rate of recurrence compared to the EP stent group (9.09% vs. 45.45%, *p* = 0.15, as shown in Table 4). Similarly, the recurrence rate was also lower in the overlapping stents group than in the LVIS stent group (9.09% vs. 27.27%, *p* = 0.59, as shown in Table 4). We found no statistically significant difference in recurrence rates between the overlapping and single-stent groups (Table 4). Additionally, there was no difference in follow-up outcomes between the overlapping and single-stent groups (Table 4).

## 4. Discussion

Studies have shown that in canines with artificially created bifurcations, pre-aneurysm changes can occur in areas where both the wall shear stress and its gradient (WSSG) exceed normal physiological levels. Such alterations, including disruptions of the internal elastic lamina and a thinned-out media, might take place before the emergence of an aneurysm [27]. To understand how the aneurysm wall responds to low WSS, a factor associated with aneurysm rupture, one must consider both biological and mechanical components. It is proposed that aneurysm rupture might directly result from low WSS in areas of the aneurysm sac [27]. While many aneurysms rupture at the tip, some break at the lateral wall of the dome or around the neck [28,29]. This makes pinpointing rupture points before surgery challenging. However, research by Keiji Fukazawa suggests that computational fluid dynamics (CFD) could identify rupture points by examining factors such as low WSS and complex flow proximity.

In a clinical context, ruptured aneurysms often have a multilobular or irregular shape, with no consistent localization of the aneurysm tip [30]. The relationship between the aneurysm and the parent vessel, correlating with the intra-aneurysmal flow pattern, has garnered research interest [31]. Baharoglu et al.’s comparative study identified the aneurysm inflow angle as a distinguishing feature of rupture status [32].

Treating dissecting aneurysms in the posterior circulation is challenging. Such aneurysms have poorer outcomes after subarachnoid hemorrhage (SAH) than those in anterior locations [2,3]. Aggressive surgical interventions, including clip reconstruction and open bypass, can be risky due to the parent vessel wall’s delicacy [1,33]. For patients with PCDAs, endovascular treatment may be preferable, especially when dealing with dissecting lesions or anatomically challenging surgical cases. Past studies have explored various endovascular treatment options such as stent-assisted coil embolization, flow diversion, and parent segment sacrifice [20,34,35]. Some dissecting aneurysms might be unsuitable for endovascular procedures because of the vessel’s size or the aneurysm’s shape, necessitating the removal of the affected segment [35]. Directly sacrificing the VA, BA segment, or PCA and PICA origins might increase cerebellar stroke risks or bilateral cerebellar hemispheric hypoperfusion [20,36,37].

Dissecting aneurysms in the posterior circulation poses unique challenges that can amplify procedural complications and risks, adversely impacting the prognosis [38,39]. The primary goal of SAH treatment is to prevent re-bleeding by isolating the aneurysm. Reconstructive endovascular therapy can maintain the parent vessel’s integrity, which might be especially beneficial for ruptured dissecting aneurysms situated in the BA, VA, PCA, or PICA origins [40,41,42]. Yet, treating these PCDAs using this method is intricate due to the necessity for immediate aneurysm occlusion and dual antiplatelet therapy amid SAH [43]. Flow diverters (FDs) have emerged as a feasible treatment for various unruptured intracranial aneurysms, especially those with large necks or complex shapes [21,44]. However, using FDs in SAH cases is controversial since they might not instantly occlude the aneurysm and could lead to potential branch occlusions, resulting in severe complications [45,46].

To the best of our knowledge, flow devices have recently been introduced with limited applications and have not yet been used to treat aneurysm ruptures in Japan [47]. Additionally, meta-analysis data suggest that FDs offer high occlusion rates and significant therapeutic effects for intracranial aneurysms. However, this method might increase the likelihood of posterior circulation ischemia, particularly with aneurysms in that area [5,45,48]. The use of acute antiplatelet agents can also elevate the risk of complications, such as aneurysmal rebleeding and external ventricular drain tract hemorrhage [43,49,50]. Using overlapping stents in conjunction with coiling might be a promising strategy for patients with ruptured dissecting aneurysms in the posterior circulation. Studies have shown that multiple overlapping stents are beneficial when using coiling techniques for shallow aneurysms [51,52].

In this study, we documented severe cases of ruptured PCDAs treated with multiple overlapping stents and coiling, ensuring the parent vessel remained intact. Liu et al. found that using multiple overlapping stents resulted in a 77.8% immediate successful embolization rate. At 21.3 months post-operation, 94.4% of patients achieved an mRS score of 0–2 [24]. Our study yielded similar results, with an immediate embolization success rate of 85.2%. In the end, 96.2% of our patients had an mRS score of 0–2 at their final follow-up. Only two patients experienced ischemic events post-operation, both of whom recovered well. No further complications were reported during the follow-up. Furthermore, the overlapping stent group displayed better outcomes and lower recurrence rates than the EP and LVIS groups. Although the clinical results were not statistically significant due to the small sample size, overlapping stents appear to have distinct advantages, making them a viable alternative technique supporting the reinserted coiling microcatheter. Even if the coiling microcatheter is reinserted after stent insertion, additional overlapping stents may positively affect coiling by fixing the jailed microcatheter. Therefore, overlapping stents may be a feasible option for acute management until the safety of flow diverters in SAH has been established, especially for patients with dissecting aneurysms.

Endovascular therapy strategies, including proximal occlusion, stent-assisted coil embolization, and stent-assisted coil angioplasty, have been employed to treat ruptured PCDAs. Overlapping stents are effective for treating fusiform and dissecting rupture aneurysms not suitable for endovascular FD devices [5,51,52]. Walsh and colleagues treated a dissecting VA aneurysm with over four overlapping stents [52]. Imahori et al. found that using five overlapping Enterprise stents is beneficial for hemorrhagic VA dissecting aneurysms [47]. Additionally, Yan and his team reported that double LVIS overlapping completely coiled seven vertebral or BA aneurysms, showing positive follow-up results [42]. These studies validate the efficacy of overlapping stents for the endovascular treatment of dissecting and fusiform aneurysms.

Our recent study revealed that overlapping different stent types, combined with coil embolization for PCDAs, is safe, effective, and has a reduced recurrence rate. In our procedures, laser-carving stents, such as Enterprise and Neuroform EZ or Altas, can bridge the aneurysms. This reduces the risk of stent trapping within the spindle aneurysm capsule, making embolization with coils more straightforward. By overlapping the LVIS, Leo, or Leo baby within the laser-carving stent, metal coverage is enhanced, leading to alterations in hemodynamics compared to using single stents for PCDAs. Overlapping stents offer greater metal coverage, potentially reducing recurrence rates and aiding in aneurysm neck reconstruction. However, there is a need to be aware of potential branch vessel occlusions. Hence, once an aneurysm is sufficiently occluded, the right dosage of antiplatelet therapy is recommended [5,49].

To sum up, our study indicates that the overlapping stents technique is effective for treating ruptured PDCAs, ensuring the parent artery remains intact. Follow-up data showed most patients experienced complete recovery with minimal health risks. Even with a sample size of 27 patients, we believe our research provides valuable insights into the potential of the overlapping stent method, positioning it as a promising treatment for ruptured PCDAs.

## 5. Study Limitations

However, the study’s retrospective nature, limited sample size, and the fact that it is a single-center study might introduce selection bias. Our findings warrant further investigations with larger sample sizes and multicenter collaborations for a comprehensive understanding of the overlapping stents’ efficacy in treating ruptured dissecting aneurysms of the posterior circulation.

## 6. Conclusions

In conclusion, our study demonstrates that overlapping stents can be used to successfully treat ruptured PDCAs, preserving the parent artery. Follow-up results showed that most patients achieved complete regression with a low risk of morbidity and mortality. Although our sample size of 27 patients may seem modest, we believe that the unique characteristics and nature of this study will contribute more data about the overlapping stent technique, making it an effective strategy for treating ruptured PCDAs.

## Figures and Tables

**Figure 1 brainsci-13-01507-f001:**
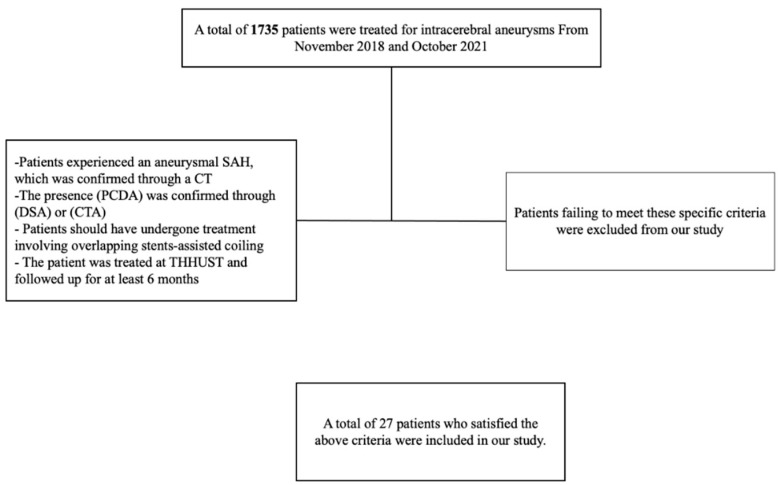
Flowchart showing the basic enrolling procedure for patients with intracerebral aneurysm in our study.

**Figure 2 brainsci-13-01507-f002:**
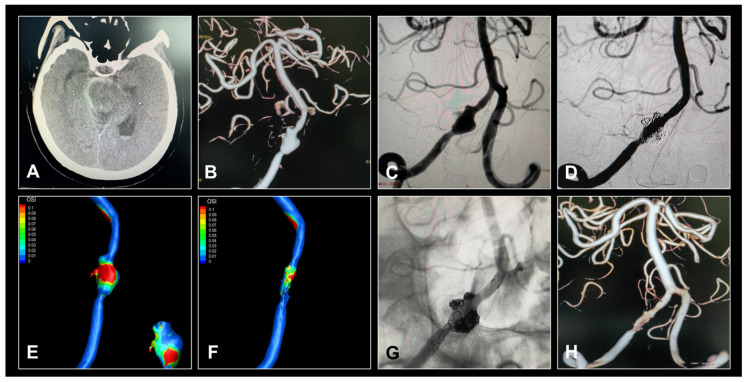
Images of a patient who had a ruptured dissecting aneurysm of the vertebral artery (VA). (**A**) Axial non-contrast-enhanced head computed tomography scan demonstrates subarachnoid hemorrhage. (**B**,**C**) Digital subtraction angiography (DSA) shows irregular fusiform dilation of the VA. (**D**) Post-treatment DSA shows an almost complete occlusion of the aneurysmal component. (**E**,**F**) Hemodynamic morphology to discriminate aneurysm rupture risk during the treatment. The average oscillatory shear index post-operation was lesser compared with that of pre-operation. (**G**,**H**) Sixteen months after the treatment, the suitable VA and three-dimensional angiography shows complete aneurysm occlusion.

**Figure 3 brainsci-13-01507-f003:**
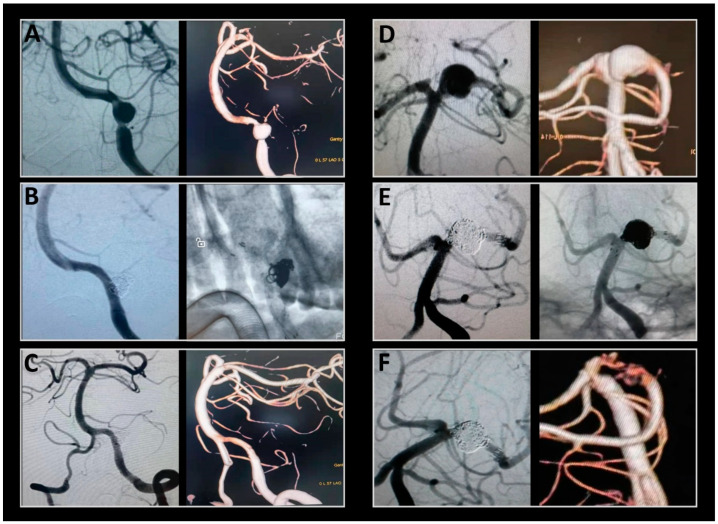
Two representative cases of a ruptured dissecting aneurysm located in the posterior circulation and treated with double overlapping stents. (**A**) Subsequent digital subtraction angiogram (DSA) shows a dissecting aneurysm of the vertebral artery (VA) in the first case. (**B**) Post-treatment DSA demonstrates almost complete occlusion of the aneurysmal component. (**C**) Twelve months after the treatment, the left VA and three-dimensional angiography show complete occlusion of the dissecting aneurysm. (**D**) The aneurysm in the working projection before treatment in the second case. (**E**) Eight coils were deployed within the aneurysmal component, and two overlapping stents were deployed subsequently. Post-treatment DSA demonstrates almost complete occlusion of the aneurysmal segment. (**F**) Twelve months after the treatment, angiography of the left PCA and three-dimensional angiography show that the aneurysms are stable and have achieved Roy–Raymond grade I embolization.

**Figure 4 brainsci-13-01507-f004:**
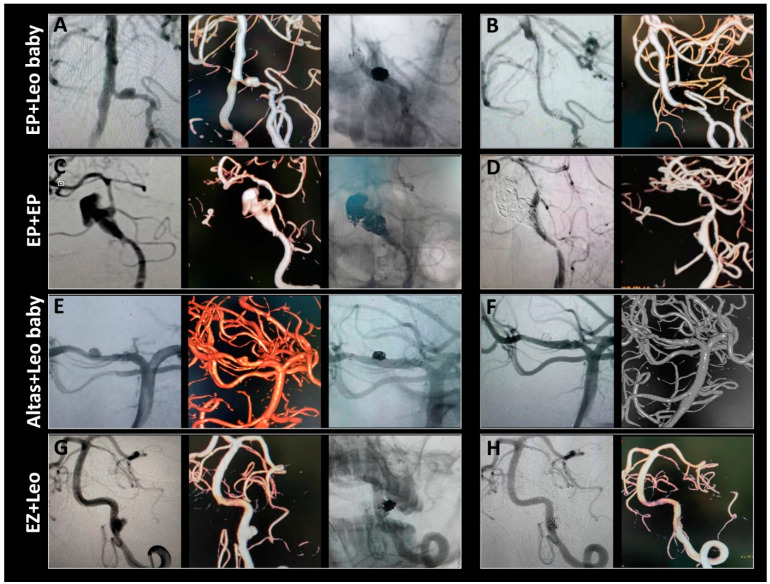
Representative cases of ruptured dissecting aneurysms located in the posterior circulation and treated with double overlapping stents. (**A**,**B**) The dissecting aneurysm in the VA was coiled and assisted with enterprise and coincided with Leo’s Baby. In the follow-up angiography image captured over 2 years later, the aneurysm shows complete occlusion. (**C**,**D**) Double enterprise stents were deployed across the aneurysm neck, and several coils were subsequently deployed within the dissecting aneurysm of the BA. The follow-up angiography was captured 8 months later, and the aneurysm shows further thrombosis. (**E**,**F**) The aneurysm in the PCA was coiled and assisted by Altas, who overlapped Leo’s Baby. Twenty-two months after the treatment, the right PCA and three-dimensional angiography shows that the aneurysms were stable and achieved Roy–Raymond grade I embolization. (**G**,**H**) The dissecting aneurysm in the VA appears coiled, assisted with Leo placed in the enterprise. Complete occlusion of the aneurysm with patency of the parent vessels on follow-up at 14 months.

**Figure 5 brainsci-13-01507-f005:**
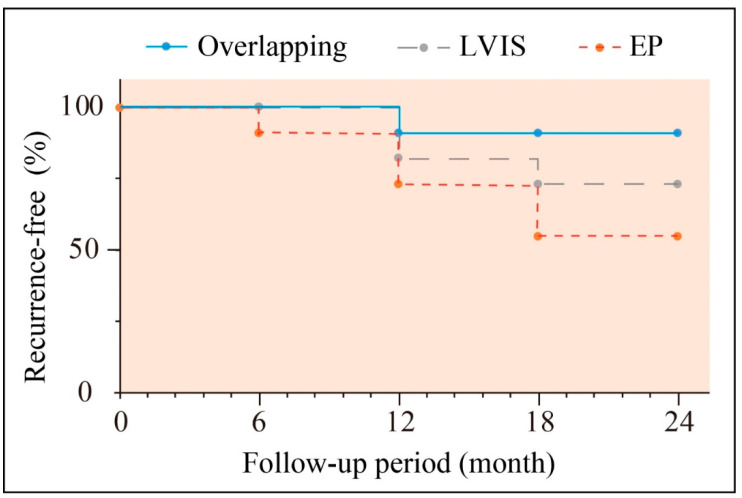
Kaplan–Meier curve of aneurysm recurrence rates during follow-up.

**Table 1 brainsci-13-01507-t001:** Clinical Data of All Patients with Overlapping stents.

Patient No.	Sex	Age	Clinical Presentation	Hunt-Hess Grade	Fisher Grade	Modified Rankin Scale	Diseased Parent Vessel	Pre-op. Ischemic Accident	Overlapping Stents	Death/ Hospice	Follow-up Time (Months)
**1**	M	56	SAH	2	2	1	VA	No	Enterprise + LVIS	No	32
**2**	F	53	SAH	3	2	2	BA	No	Nruroform EZ + Leo	No	9
**3**	M	38	SAH	1	2	1	VA	No	Enterprise + LVIS	No	17
**4**	F	57	SAH	1	1	5	PCA	No	Enterprise + LVIS	No	15
**5**	F	9	SAH	4	4	5	BA	No	Enterprise + Enterprise	No	33
**6**	M	49	SAH	3	2	1	BA	No	Enterprise + LVIS	No	19
**7**	M	10	SAH	4	4	2	PCA	No	Nruroform Atlas + Leo baby	No	22
**8**	M	60	SAH	1	2	2	VA	No	Enterprise + Leo baby	No	28
**9**	M	58	SAH	3	3	3	BA	No	Enterprise + LVIS	No	7
**10**	M	53	SAH	1	2	3	PICA	No	NruroformAtlas + Enterprise	No	11
**11**	M	48	SAH	2	2	1	VA	No	Enterprise + Leo	No	13
**12**	M	57	SAH	1	1	2	BA	No	Enterprise + LVIS	No	20
**13**	F	58	SAH	1	2	2	VA	No	Enterprise + Leo	No	19
**14**	M	63	SAH	3	3	2	VA	No	Enterprise + LVIS	No	10
**15**	M	58	SAH	2	2	5	BA	No	Enterprise + Enterprise	No	8
**16**	M	51	SAH	5	4	5	VA	No	Nruroform EZ + Leo	Yes	/
**17**	F	67	SAH	1	2	3	VA	No	Enterprise + Leo	No	12
**18**	M	57	SAH	2	2	2	VA	No	Enterprise + LVIS	No	16
**19**	F	49	SAH	5	4	3	vA	Yes	Enterprise + LVIS	No	6
**20**	M	55	SAH	3	2	2	VA	No	Nruroform EZ + Leo	No	14
**21**	M	63	SAH	2	2	4	VA	No	Enterprise + LVIS	No	18
**22**	M	37	SAH	2	2	2	PCA	No	Enterprise + Enterprise	No	12
**23**	F	64	SAH	1	2	3	VA	No	Enterprise + LVIS	No	23
**24**	F	77	SAH	2	2	5	BA	No	Enterprise + Enterprise	No	12
**25**	M	51	SAH	1	2	3	BA	No	Enterprise + Leo baby	No	21
**26**	M	57	SAH	2	2	2	PICA	No	Enterprise + Leo baby	No	9
**27**	F	59	SAH	4	1	4	VA	No	Nruroform EZ + Leo	No	15

SAH, subarachnoid hemorrhage. VA, Vertebral artery; BA, Basilar artery; PCA, Posterior cerebral artery; PICA, Posterior inferior cerebral artery.

**Table 2 brainsci-13-01507-t002:** Aneurysm Clinical and precedure-related comlications.

Variable (*n* = 27)	Value (%)
**Size of aneurysms**	
**Small (<10 mm)**	16 (59.3%)
**Large (10~25 mm)**	8 (29.6%)
**Giant (>25 mm)**	3 (11.1%)
**Overlapping stents**	
**Enterprise + Enterprise**	4 (14.8%)
**Enterprise + LVIS**	11 (40.7%)
**Enterprise + Leo/Leo baby**	6 (22.2%)
**Neuroform EZ/Altas + Enterprise**	1 (3.7%)
**Neuroform EZ/Altas + Leo/Leo baby**	5 (18.5%)
**Complication**	
**Assess site hematoma requiring treatment**	0 (0%)
**Acute re-hemorrhage**	0 (0%)
**Cerebral edema**	5 (18.5%)
**Hydrocephalus**	7 (25.9%)
**Symptomatic procedure-related Stroke**	2 (7.4%)
**Mass effect**	0 (0%)
**Death/Hospice**	1 (3.7%)

Values are presented as number (%) of aneurysms or mean ± SD.

**Table 3 brainsci-13-01507-t003:** Clinical outcomes and radiological characteristics.

Characteristics	Immediately (*n* = 27)	Discharge (*n* = 26)	Follow-Up (*n* = 26)
**A** **ngiography**			
**Raymond Grade I**	16 (59.3%)	/	24 (92.3%)
**Raymond Grade II**	7 (25.9%)	/	2 (7.7%)
**Raymond Grade III**	4 (14.8%)	/	0 (0%)
**Modified Rankin Scale**			
**mRS 0-2**	14 (51.9%)	19 (73.1%)	25 (96.2%)
**mRS 3-4**	8 (29.6%)	6 (23.1%)	1 (3.8%)
**mRS 5-6**	5 (18.5%)	1 (3.8%)	0 (0%)

Values are presented as number(%)of aneurysms or mean ± SD. mRS, modified Rankin Scale.

**Table 4 brainsci-13-01507-t004:** Comparison of aneurysm characteristics, baseline clinical data, and outcomes between the overlapping group and the EP group and between the overlapping group and the LVIS group.

			Overlapping Group *n* = 11	EP Group *n* = 11	*p*-Value	Overlapping Group *n* = 11	LVIS Group *n* = 11	*p*-Value
Age, years mean ± SD			55.54 ± 7.67	53.63 ± 3.70	0.47	55.54 ± 7.67	54.36 ± 5.14	0.68
Male sex, n (%)			8 (72.7)	7 (63.6)	1.00	8 (72.7)	7 (63.6)	1.00
Hypertension n (%)			7 (63.6)	7 (63.6)	1.00	7 (63.6)	8 (72.7)	1.00
Diabetes mellitus, n (%)			6 (54.5)	6 (54.5)	1.00	6 (54.5)	7 (63.6)	1.00
Hyperlipidemia n (%)			5 (45.5)	6 (54.5)	0.67	5 (45.5)	5 (45.5)	1.00
Smoking n (%)			7 (63.6)	6 (54.5)	1.00	7 (63.6)	6 (54.5)	1.00
Alcohol n (%)			6 (54.5)	6 (54.5)	1.00	6 (54.5)	7 (63.6)	1.00
Hunt–Hess grade					1.00			0.70
		1–2, n (%)	7 (63.6)	6 (54.5)		7 (63.6)	5 (45.5)	
		3–5, n (%)	4 (36.4)	5 (45.5)		4 (36.4)	6 (54.5)	
Fisher grade					1.00			1.00
		1–2, n (%)	8 (72.7)	7 (63.6)		8 (72.7)	7 (63.6)	
		3–4, n (%)	3 (27.3)	4 (36.4)		3 (27.3)	4 (36.4)	
Size (mm)			8.62 ± 1.35	8.09 ± 1.11	0.32	8.63 ± 1.35	8.37 ± 0.94	0.61
Neck (mm)			8.78 ± 1.41	8.45 ± 1.11	0.54	8.78 ± 1.41	8.56 ± 1.00	0.68
Results								
	Angiographic outcome				0.15			0.59
		No-recurrence, n (%)	10 (90.91)	6 (54.55)		10 (90.91)	8 (72.73)	
		Recurrence, n (%)	1 (9.09)	5 (45.45)		1 (9.09)	3 (27.27)	
	mRS score at last follow-up				0.59			1.00
		0–2, n (%)	10 (90.91)	8 (72.73)		10 (90.91)	9 (81.82)	
		3–6, n (%)	1 (9.09)	3 (27.27)		1 (9.09)	2 (18.18)	

## Data Availability

The raw data supporting the conclusions of this article will be made available by the authors without undue reservation.

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
