# Peer review of "Overlapping Stent Treatment for Ruptured Dissecting Aneurysms in Posterior Circulation"

_brainsci, 2023, doi:10.3390/brainsci13111507_

Round 1
Reviewer 1 Report
Dear Authors,
Thank you for presenting a case series of especially challenging situation of ruptured aneurysm in posterior circulation. The results are promising but I believe that you should underline your results with following changes in your paper.
1. You have written: "Recently, overlapping stents have been tested to treat PCDAs, and it was shown favorable conditions for blood flow to increase thrombosis". THere is no citation here and it would be useful.
2. Do you have any own historical data upon effectiveness of therapy of ruptured aneurysms in posterior circulation with other modalites? If so - please provide them to make any comparisons possible
3. Showing data from other publications about results (e.g., mortality, final Rankin) of therapy in this condition (with the same or alternative methods) would allow the reader to see significance of your results
4. Please, provide some data about the one patient who was transferred to hospice - what happened? Did the procedure in that case go wrong?
English language requires moderate improvement.
Author Response
Thank you, we have all the information in the file

Reviewer 2 Report
Dear Authors. Thank you for your valuable efforts and the interesting article. Here are my comments:
1. Second paragraph of introduction should be more clearly discussed for example explaining the parent and patent arteries. (L40-45)
2. In methodology, telephone follow up is not an optimal method.
3. The baseline mRS measurement is not mentioned
4. Line 108-110: lack of anticoagulation is previously mentioned
5. The Roy-Raymond scale should be explained better
6. It is better to clarify the reason for hospice transfer of a patient
7. It should be mentioned that complications such as hydrocephalus are probably non-procedural dependent.
8. The sentence in lines 221-222 is not quite understandable for me. Then next sentence is also vague and not correctly presented.
9. Moderate English language editing is needed, especially in the discussion section.
10. The technique of using overlapping stents should be discussed and introduced better to the readers.
11. Limitations of the study should be mentioned, if encountered
Quality of english language was acceptable overally, but there were some cases of grammatical mistakes and some sentences needs to be restructured to be presented better.
Author Response
Thank you, we have included all the information in the file.

Reviewer 3 Report
The unfavorable prognosis associated with ruptured dissecting aneurysms originating from the posterior intracranial circulation, along with the challenging treatment landscape, makes this topic intriguing for readers. In their study, the authors utilized a cohort of 27 patients to establish the foundation for their investigation. However, this population maybe is not large enough for decision-making. They meticulously measured a range of valuable parameters from these patients, underscoring the robustness of their approach. However, the current statistical analysis falls short of substantiating the primary conclusion of the paper. Consequently, I anticipate an improved and revised version of this paper that addresses these concerns.
Comments:
1. "We describe the clinical presentation, radiologic features, endovascular procedure, and clinical outcomes of 27 consecutive patients with these lesions, treated using overlapping stent placement". However, this cannot be a part of the background in the abstract section.
2. The numerical results of this paper are not mentioned in the abstract section. The results section of the abstract only describes the database.
3. The introduction section is incomplete. Some studies have highlighted and clarified the effects of bifurcations in the aneurysm, the morphology of the aneurysm neck, and the impact of hemodynamic changes in small-sized aneurysms on rupture risk and treatment [10.1134/S0021894417060025] and [10.1007/s10143-020-01367-3]. It is necessary to address these points in the introduction section.
4. The last paragraph of the introduction section is not clear enough for readers. You need to further clarify your hypothesis and outline the tools you will use to test its validity.
5. Referring to references for further details on the 'Hunt-Hess' grade at clinical presentation and the 'Fisher' grade on the initial non-contrast head CT is recommended.
6. Making decisions about papers with a follow-up of a couple of months requires survival analysis and Kaplan-Meier curves.
7. There are many more important morphometric parameters related to aneurysms. Why did you only report the size and percentage of aneurysm sac obliteration? Do you believe these are of great importance? If so, please explain.
8. Your results do not fully support the main finding of this paper, 'Overlapping stents may become an effective strategy for treating ruptured PCDAs.' A stronger statistical analysis is required.
9. Previous studies have highlighted the role of blood dynamics and wall shear stress (WSS) in assessing rupture risk of aneurysms [10.1016/j.jocn.2011.02.014] and [10.1016/j.wneu.2013.02.012] and [10.1016/j.medengphy.2007.04.011]. It is necessary to discuss their potential effects on your results and recommend them for future studies.
10. There are many effective parameters influencing rupture risk that you have not considered in your evaluation. How did factors like family history, smoking, etc. [10.1155/2012/271582] and [10.3171/jns.2002.96.1.0003] affect your findings?
11. There is a lack of robust numerical analysis in this paper. While you worked with numerous numerical parameters, it's unclear how you arrived at these findings without proper statistical analysis. Many sections of this paper contain sentences burdened with statistical references but lack corresponding analyses.
12. Do you think 27 patients are enough to make the decision and mention this big claim that you mentioned in the summary section?
13. English language level of this paper is not enough for publishing this paper in this journal.
Moderate level of English language
Author Response
Thank you, we have included all in the information in the vile

Round 2
Reviewer 1 Report
Dear Authors,
THank You for addressing my suggestions. I have no further remarks.
Author Response
Comments and Suggestions for Authors
Dear Authors,
THank You for addressing my suggestions. I have no further remarks.
Response: Thank you for your suggestions. We're very grateful to you.

Reviewer 3 Report
Thank you for the corrections. I cannot find the answers to my comments in the revised files.
Comment 3:... It is necessary to address these points in the introduction section. But I cannot find them in Line 79-102 that you mentioned in your response. Also Comment 4, I talked about the hypothesis and outline but you said that you applied them in Line 124-128 BUT this is the Method section?! also there is a similar issue for Comments 1, 2, 9, and 10.
PLEASE HIGHLIGHT THE SPECIFIC SENTENCES IN THE MAIN MANUSCRIPT THAT YOU CORRECTED OR CHANGED IN THE REVISED VERSION OF THE PAPER.
Comment 5: I couldn't find the supplementary materials.
Comment 6: I can only find 3 figures in the manuscript
Comment 8: Where is the '...conduct a more robust statistical analysis to ensure that our findings ...' in the main manuscript?
Comment 11: We will .... and we are committed to ...
OK, where are the corrections? where is the numerical analysis in the revised version?
Comment 12: While our sample size of 27 patients may seem modest, we believe that the unique characteristics and nature of our study population provide valuable insights into the topic at hand. ... will continue to address any concerns raised during the peer-review process.
I cannot understand what "provide valuable insights into the topic at hand" or "will continue to address any concerns raised during the peer-review process" means. Did you correct the manuscript? Where? what changes?
Not bad
Author Response
Please see the attachmen

Round 3
Reviewer 3 Report
This paper can be published